# Mitochondrial DNA and Electron Transport Chain Protein Levels Are Altered in Peripheral Nerve Tissues from Donors with HIV Sensory Neuropathy: A Pilot Study

**DOI:** 10.3390/ijms25094732

**Published:** 2024-04-26

**Authors:** Ali Boustani, Jacqueline R. Kulbe, Mohammadsobhan Sheikh Andalibi, Josué Pérez-Santiago, Sanjay R. Mehta, Ronald J. Ellis, Jerel Adam Fields

**Affiliations:** 1Department of Psychiatry, University of California San Diego, San Diego, CA 92093, USA; aboustani@health.ucsd.edu (A.B.);; 2Department of Neurosciences, University of California San Diego, San Diego, CA 92093, USA; mandalibi@health.ucsd.edu (M.S.A.);; 3Division of Cancer Biology, University of Puerto Rico Comprehensive Cancer Center, San Juan, PR 00927, USA; 4Department of Medicine, University of California San Diego, San Diego, CA 92093, USA

**Keywords:** DSP, distal symmetric polyneuropathy, mtDNA, common deletion mutation

## Abstract

Distal sensory polyneuropathy (DSP) and distal neuropathic pain (DNP) remain significant challenges for older people with HIV (PWH), necessitating enhanced clinical attention. HIV and certain antiretroviral therapies (ARTs) can compromise mitochondrial function and impact mitochondrial DNA (mtDNA) replication, which is linked to DSP in ART-treated PWH. This study investigated mtDNA, mitochondrial fission and fusion proteins, and mitochondrial electron transport chain protein changes in the dorsal root ganglions (DRGs) and sural nerves (SuNs) of 11 autopsied PWH. In antemortem standardized assessments, six had no or one sign of DSP, while five exhibited two or more DSP signs. Digital droplet polymerase chain reaction was used to measure mtDNA quantity and the common deletions in isolated DNA. We found lower mtDNA copy numbers in DSP+ donors. SuNs exhibited a higher proportion of mtDNA common deletion than DRGs in both groups. Mitochondrial electron transport chain (ETC) proteins were altered in the DRGs of DSP+ compared to DSP− donors, particularly Complex I. These findings suggest that reduced mtDNA quantity and increased common deletion abundance may contribute to DSP in PWH, indicating diminished mitochondrial activity in the sensory neurons. Accumulated ETC proteins in the DRG imply impaired mitochondrial transport to the sensory neuron’s distal portion. Identifying molecules to safeguard mitochondrial integrity could aid in treating or preventing HIV-associated peripheral neuropathy.

## 1. Introduction

The life expectancy of people with HIV (PWH) has significantly increased in the post-antiretroviral therapies (ARTs) era. Advancements in ARTs have revolutionized the landscape for PWH, prolonging lifespans and improving overall health [1,2]. However, this has given rise to a new set of challenges in the form of chronic comorbidities of HIV infection, such as chronic inflammation, dyslipidemia, and insulin resistance, potentially leading to cardiovascular complications and neuronal injuries including peripheral neuropathies [3,4]. HIV-associated distal sensory polyneuropathy (HIV DSP) is a common complication of HIV infection, with prevalence rates ranging from 10% to 45% in PWH [5,6,7]. DSP symptoms include pain, decreased sensation, allodynia, and paresthesia [8,9].

Neurons depend on ATP for axonal transport and maintaining ionic gradients, which are crucial for generating action potentials and facilitating synaptic activity [10]. Both ART and HIV proteins (ex: gp120) have been implicated in HIV-DSP [9,11], likely through mechanisms that involve mitochondrial function [12,13,14]. HIV infection impairs mitochondrial respiration, electron transport chain (ETC) activity, and mitochondrial DNA (mtDNA) replication [15,16]. ART induces mitochondrial toxicity, impairs oxidative phosphorylation, increases reactive oxygen species, reduces ATP synthesis, and alters mitochondrial biogenesis [17], effects which have been demonstrated to occur in sensory neurons and have been implicated in antiretroviral toxic neuropathy (ATN) and peripheral neuropathy [17,18,19]. Further, HIV and ART have been implicated in the inhibition of the gamma DNA polymerase, the enzyme responsible for mtDNA replication [20], compromising the mitochondrial genome and leading to mitochondrial dysfunction [21].

Therefore, mtDNA damage has emerged as a focal point of investigation in understanding the mechanisms underlying neuropathy in PWH undergoing ART [22]. The specific metric utilized to gauge mtDNA damage in this context is the common deletion of a molecular marker indicative of genetic alterations within the mitochondrial genome [23]. The mtDNA common deletion is a 4977-base pair deletion that has been found in increasing abundance in older age in human tissue obtained from multiple sites, including the CNS, heart, liver, kidney, and skeletal muscles [24,25,26]. Common deletion mutations are somatic mutations. Within cells, common deletion mutations exist in heteroplasmy, along with wild-type mitochondria. The proportion of mutated mtDNA relative to normal mtDNA can vary, leading to varying degrees of mitochondrial dysfunction depending on this ratio. Wild-type mitochondria are able to compensate for biochemical deficits attributable to common deletion mitochondria up to a certain threshold, beyond which pathologic phenotypes emerge. Therefore, cells with high levels of the common deletion mutation should exhibit more pronounced mitochondrial dysfunction compared to cells with lower levels of this mutation [27,28,29]. In a noteworthy prior study of 67 HIV-positive individuals undergoing ART, there was an inverse correlation between mtDNA deletions and peripheral neuropathy as assessed by intraepidermal nerve fiber density and sural nerve amplitude, further suggesting a potential role of mitochondrial dysfunction in the neuropathic processes associated with HIV and its treatment [30,31].

Axonal mitochondria are assembled in the neuronal cell body and transported down the length of axons utilizing microtubule-based machinery, eventually becoming anchored in the axon according to energy demand [32]. Therefore, mitochondria located in the distal axons of long peripheral nerves, such as the sural nerve (SuN), are at an increased risk of accumulating mtDNA mutations [16,30,33] and are heavily reliant on quality control measures, including fission and fusion [34]. These are processes that can be impaired by neuroinflammatory processes such as HIV [35,36,37].

While the mechanisms underlying these phenomena remain under investigation, the recognition of HIV-mediated mitochondrial impairment has important implications for both the understanding of HIV pathogenesis and the development of therapeutic strategies. The complex interplay between the virus, treatment modalities, and individual factors underscores the importance of a comprehensive approach to healthcare for PWH, addressing not only viral suppression but also monitoring and controlling the long-term conditions that can arise alongside prolonged ART use, such as peripheral neuropathy. Further elucidating the molecular pathways involved in HIV-induced mitochondrial dysfunction may unveil novel targets for intervention, not only in the context of HIV treatment but also in addressing broader mitochondrial-related diseases.

Therefore, the overall aim of this study was to compare the differences in mitochondrial mutations (common deletion number), nuclear-encoded fission and fusion proteins (MFN1 (mitofusion 1), MFN2, and dynamin-related protein 1 (DRP1)), and mtDNA-encoded ETC proteins (ATP-synthase, Complex I, II, III, and IV) in PWH with DSP (DSP+) and without DSP (DSP−) and between dorsal root ganglion (DRG) and SuN samples. We hypothesized that we would see increased mitochondrial mutations and decreased fission, fusion, and ETC protein concentrations in DSP+ individuals compared to DSP− individuals and that these effects would be more pronounced in the SuN.

## 2. Results

Postmortem DRG and SuN specimens of 11 PWH were studied in this research. The mean age of the sample at the time of death was 44 years, and 27.2% were female (Table 1). Their last median CD4 cell count was 12 (7–50.5) cells/mm^3^, and their nadir CD4 cell count was 9.5 (2–23) cells/mm^3^. Eight (72.7%) were on ART at the time of death. Five (45.5%) PWH met the definition of DSP. No significant differences were observed between DSP+ and DSP− patients in terms of demographic and clinical characteristics, including age, gender, most recent and nadir CD4 cell counts, frequency and duration of ART and regimen, non-nucleoside analog reverse transcriptase inhibitors (NNRTIs), or highly active ARTs (HAART). The differences in postmortem intervals (PMIs) between the DSP− (mean = 17.2 h, standard deviation = 15.2 h) and DSP+ (mean = 24 h, standard deviation = 4 h) groups were not significant.

### 2.1. mtDNA Copy Number and Common Deletion Relative Abundance (RA) in DSP

There was a significant reduction in mtDNA copy number per cell in the DRGs of DSP+ decedents compared to DSP− decedents and a non-significant reduction in mtDNA copy number per cell in the SuNs of DSP+ decedents compared to DSP− decedents (Figure 1a). There was a non-significant increase in mtDNA copy number per cell in SuNs compared to DRGs in both DSP− and DSP+ decedents (Figure 1a). mtDNA common deletion was normalized to mtDNA copy number. mtDNA common deletion was not significantly different between the groups (Figure 1b). However, there was a trend for mtDNA common deletion to be elevated in the DRGs of DSP+ decedents compared to DSP− decedents and for mtDNA common deletion to be elevated in the SuN compared to the DRG in both groups (Figure 1b). Of note, one SuN specimen, collected from a DSP− decedent who was diagnosed with HIV six months prior to death, showed a markedly higher level of mtDNA common deletion relative abundance (RA) (approximately 100 times more) compared to other mtDNA common deletion RA levels from this group. The removal of this data point resulted in significantly increased mtDNA common deletion in the SuNs of DSP+ compared to DSP− decedents (Appendix A).

### 2.2. Mitochondrial Dynamics and ETC Protein Expression Levels Are Altered in DSP

The immunoblots of DRG and SuN lysates from DSP− and DSP+ decedents revealed bands corresponding to MFN1, MFN2, DRP1, and ACTB (β-actin) (Figure 2a). Densitometry analyses revealed no significant difference in levels of MFN1 (Figure 2b) or MFN2 (Figure 2c) between any of the groups. Densitometry analyses of the band corresponding to DRP1 revealed a ~40% increase in DRP1 levels in the DRGs of DSP+ decedents compared to DSP− decedents (Figure 2d; ** *p* < 0.01). The immunoblots of DRG and SuN lysates from DSP− and DSP+ decedents revealed bands corresponding to ATP synthase, Complex III, Complex II, Complex IV, Complex I, p-DRP-1, and ACTB (Figure 2e). Densitometry analyses of the band corresponding to ATP synthase revealed a significant increase in its levels in the SuNs of DSP− decedents compared to the ATP synthase levels in their DRGs (Figure 2f; * *p* < 0.05). Densitometry analyses of the band corresponding to Complex III revealed a significant increase in the SuNs of DSP− decedents compared to their DRGs (Figure 2g; ** *p* < 0.01) and the SuNs of DSP− decedents compared to DSP+ decedents (Figure 2g; ** *p* < 0.03). Densitometry analyses of the band corresponding to Complex II revealed a significant increase in the DRGs of DSP+ decedents compared to DSP− decedents (Figure 2h; * *p* < 0.05). Densitometry analyses of the band corresponding to Complex IV revealed a significant increase in the SuNs of DSP− decedents compared to their DRGs (Figure 2i; * *p* < 0.05). Densitometry analyses of the band corresponding to Complex I revealed a significant increase in the DRGs of DSP+ decedents compared to the DRGs of DSP− decedents (Figure 2j; * *p* < 0.05) and a significant increase in the SuNs of DSP+ decedents compared to the SuNs of DSP− decedents (Figure 2j; * *p* <0.05). Densitometry analyses of the band corresponding to p-DRP-1 demonstrated an increase in the DRGs of DSP− decedents compared to their SuNs (Figure 2k; * *p* < 0.05) and compared to the DRGs of DSP+ decedents (Figure 2k; * *p* < 0.05). Therefore, complexes I, II, and III increased in the DRG of DSP+ PHW while only complexes I and II increased in the SuN of DSP+ PHW and complex III decreased (Figure 3). Original blot images are included (Appendix A).

## 3. Discussion

This study used ddPCR to analyze mtDNA and immunoblots to assess the expression of mitochondrial dynamics and ETC proteins in postmortem DRG and SuN specimens from individuals with and without DSP. These methods allowed for a comprehensive exploration of mitochondrial dynamics and function in the context of sensory neuropathy in PWH. We found a significant reduction in mtDNA copy number per cell in both the DRGs and SuNs of individuals with DSP. This observation suggests a link between HIV infection, ART, mitochondrial dysfunction, and the pathogenesis of sensory neuropathy. The reduction in mtDNA copy number suggests that mitochondrial dysfunction may result from or cause neurodegenerative processes in the context of DSP.

Previous studies link HIV infection to DSP as a result of the virus itself or via toxic neuropathy associated with certain D-drugs (dideoxynucleoside analog reverse-transcriptase inhibitors), specifically stavudine, didanosine, and zalcitabine [12], or, most likely, both together. Ten participants had prior exposure to the D-drugs. mtDNA common deletion RA increased in the SuN compared to the DRG in both DSP+ and DSP− groups, suggesting a potential cumulative effect of mtDNA common deletion on the pathogenesis of DSP, which predominantly affects the distal parts of the nerve fibers compared to the DRG. This finding supports the notion that mitochondrial genome damage, as reflected by the common deletion, may play a role in the development and progression of DSP in long nerves. The specificity of this alteration in distal nerves highlights the importance of investigating regional differences in mitochondrial dynamics within the peripheral nervous system.

Our study extended to an analysis of ETC proteins, finding a significant elevation in the levels of these proteins within the DRGs of DSP+ donors, particularly in Complex I and DRP-1. NRTIs cause a notable decrease in the activity of Complex IV and a targeted blockage of Complex I [38]. In this study, 91.9% (10/11) of the donors had a history of NRTIs, specifically D-drugs. No differences were observed in the expression of ETC Complex IV between DSP+ and DSP− samples. However, there was an upregulation of Complex IV in the SuN compared to DRG in DSP− donors. Lehmann et al. reported a decrease in the expression of subunits of Complex IV in SuN samples from DSP+ PWH compared to both DSP− individuals and controls [16]. This regional disparity in ETC protein expression suggests a compensatory mechanism in response to mitochondrial dysfunction. The upregulation of ETC proteins may indicate an attempt to overcome impaired mitochondrial function by enhancing the production of key components involved in energy production. However, the lack of a similar upregulation in the SuN suggests potential limitations in the transport of mitochondria to the distal portions of sensory neurons, possibly contributing to the observed neuropathic symptoms [39].

Several studies have highlighted the involvement of DRP-1 in neuroinflammation and DSP through multiple pathways [39,40,41]. Research has demonstrated that the spinal intrathecal administration of oligodeoxynucleotide (ODN) antisense to Drp1 reduces the expression level of Drp-1 in primary afferent fibers, reducing ddC-induced mechanical hyperalgesia in male Sprague Dawley rats. Furthermore, this study revealed that the intradermal injection of Mitochondrial Division Inhibitor 1 (mdivi-1), a selective inhibitor of Drp1-dependent mitochondrial fission, significantly alleviated mechanical hyperalgesia induced by ROS and ddC [41]. The current study found that DRP-1 expression within the DRG was markedly higher in DSP+ donors than DSP− donors. ATP synthase is the complex assembled in the mitochondria and transported to the cell surface by DRP-1 [42]. We found that ATP synthase exhibited a remarkable upregulation in the SuN compared to the DRG in donors without DSP.

A prior study showed macrophage activation and infiltration in the DRG of PWH with DSP [43]. Macrophage activation contributes to mitochondrial dysfunction by altering ETC activity and the TCA cycle [44]. Activated macrophages also induce an upregulation of glucose and glutamine utilization and a shift toward anabolic pathways. These interactions between macrophage activation and mitochondrial function are bidirectional. Thus, itaconate, produced in the mitochondrial matrix by the TCA cycle metabolite cis-aconitate, regulates multiple aspects of macrophage function.

The strength of this work lies in its comprehensive investigation of mitochondrial dynamics and ETC protein expression in the context of distal sensory polyneuropathy (DSP) in people living with HIV. Prior research in this area has often focused on individual aspects of mitochondrial function or neuropathy, but this study integrates multiple facets to provide a better understanding of the relationship between mitochondrial dysfunction and DSP. Additionally, the inclusion of postmortem DRG and SuN specimens from a well-characterized population of people living with HIV enhances the clinical relevance and applicability of the findings.

Our study has several noteworthy limitations. The relatively small sample size due to limited access to autopsied DRG and SuN specimens may affect the generalizability of our findings. Further, this study lacks specimens from HIV-/DSP− controls to establish a baseline for the parameters measured. We aim to complete future studies with a larger number of specimens from a diverse group of decedents including HIV-/DSP− and HIV-/DSP+ decedents. While the biochemical and molecular analyses of DRG and SuN samples are novel in the context of DSP in PWH, histopathologic assessments of the specimens are needed to confirm that neuropathological changes are consistent with the protein and mtDNA alterations described here. It is important to include such assessments in future studies. The PMIs for the specimens included in this study are higher than ideal for neuropathological assessments, and effort will be made in the future to examine tissues with PMIs under 8 h. However, the differences in PMI between the groups are not significant, and the changes in mtDNA and protein levels between the groups are not reflective of increased degradation in the group with the larger PMI.

Future research with larger cohorts is warranted to validate and expand upon our results. Additionally, it is not completely known to what extent the virus itself or ART contribute to the pathogenesis of DSP separately. The impact of ART on mitochondrial function varies widely depending on the specific drugs used, their dosage, and individual patient factors [45]. Therefore, investigating the functional consequences of the observed molecular alterations and the direct impact of specific ARTs on mitochondrial function could provide a more comprehensive understanding of the underlying mechanisms. Future studies could also explore the potential use of mitochondrial protective agents or interventions to enhance mitochondrial transport as therapeutic strategies for DSP. Longitudinal studies tracking mitochondrial dynamics in PWH over time and in response to different antiretroviral regimens could further elucidate the progressive nature of mitochondrial dysfunction in the context of HIV. Finally, as the risk of mitochondrial toxicity from newer ART regimens decreases, the more significant mitochondrial impacts on PWH moving forward may be direct viral toxicity in combination with metabolic disease. Future studies could also explore the potential use of mitochondrial protective agents or interventions to enhance mitochondrial transport as therapeutic strategies for DSP. Targeting pathways that facilitate the transport of mitochondria to distal nerve endings may represent a therapeutic approach to address the specific neuropathic symptoms observed in DSP. Longitudinal studies tracking mitochondrial dynamics in PWH over time and in response to different antiretroviral regimens could further elucidate the progressive nature of mitochondrial dysfunction in the context of HIV. The regional differences in ETC protein expression identified here point to new avenues for research into the specific mechanisms by which mitochondrial dysfunction contributes to sensory neuropathy and may lead to identifying novel drug targets for intervention. Additionally, research into the potential interplay between mitochondrial health, immune function, and neuroinflammation in the context of DSP could provide valuable insight into the multiple causes of this condition and inform the development of comprehensive treatment strategies.

## 4. Materials and Methods

### 4.1. Study Population

PWH were evaluated at the University of California, San Diego, in the HIV Neurobehavioral Research Center and the California NeuroAIDS Tissue Network. An IRB approved this research with the approval code #080323, and each participant gave informed consent. Data were collected according to a protocol of comprehensive neuromedical, neurobehavioral, and laboratory assessments that were standardized across sites. DRG and SuN specimens were obtained from this cohort posthumously and processed as described below.

### 4.2. Phenotype Definitions

HIV-associated DSP was determined using the CHARTER study protocol which has previously been described in detail [46]. Briefly, clinicians trained in HIV neurological disorders performed a standardized, targeted neurological examination to evaluate DSP signs, including diminished ability to recognize vibrations, reduced sharp–dull discrimination in the feet and toes, and reduced ankle reflexes. Neuropathy symptoms were also assessed in the legs, feet, and toes, including bilateral neuropathic pain and dysesthesias (burning, aching, or shooting), paresthesia, and loss of sensation. Using a standardized form and a structured interview, clinicians classified neuropathic pain into the following 5 severity levels: none, slight (occasional, fleeting), mild (frequent), moderate (frequent, disabling), and severe (constant, daily, disabling, requiring analgesic medication or other treatment). A comprehensive evaluation assessing limb strength and sensory and motor symptoms was also conducted. The presence of at least 2 signs bilaterally was considered to be evidence of DSP [47]. Based on this evaluation, the cohort was categorized into two groups: individuals without neuropathy (DSP−) and those with neuropathy (DSP+).

### 4.3. Quantification of mtDNA and the Proportion of Mitochondria Carrying the Common Deletion

DNA quantification was performed using the highly sensitive droplet digital PCR platform [48]. Genomic DNA was extracted from DRG and SuN specimens and then fragmented using three different methods:

Enzymatic digestion—Extracted DNA was enzymatically digested using the BamHI enzyme (ThermoFisher Scientific, New York, NY, USA). Then, 250 ng DNA in 10 μL 10 mM Tris-EDTA (TE) buffer was added to 10 μL 0.2× BamHI buffer containing 10 U BamHI enzyme and incubated at 37 °C for 1 h.

Sonication—Sonication was performed by adding 375 ng DNA in 30 μL TE buffer to a microtube. A range of sonication target lengths (200 bp, 500 bp, 800 bp, 2 kb, and 5 kb) were tested using a M220 Focused Ultrasonicator, Covaris, Woburn, MA, USA. The quantification of target sequences using digital droplet polymerase chain reaction (ddPCR) was used to determine which sonication length was optimal.

QIAshredder spin columns—An amount of 375 ng of DNA in 30 μL TE buffer was introduced into the QIAshredder spin column and centrifuged for 2 min at 13,000 rpm.

The resulting fragmented DNA concentration was 12.5 ng/μL regardless of the fragmentation method used. Fragmented DNA was then introduced into the ddPCR reaction.

The mtDNA copy number was measured by targeting the mitochondrial NADH dehydrogenase 2 (ND2), while the mtDNA common deletion was measured using a primer–probe set targeting the end of the deletion. Ribonuclease P protein subunit p30 (RPP30) was used as a cellular control, as 2 copies are present in each cell. Common deletion and RPP30 assays were multiplexed using 50 ng of DNA per replicate, while the ND2 assay was performed alone using 50 pg of DNA per replicate. Both assays were run in triplicate [49]. Quantification was performed as follows: 50 ng or 50 pg of DNA (in 4 μL) was added to a master mix consisting of 10 μL of 2× Bio-Rad supermix for probes, 1 μL of 20× Primer/FAM-ZEN common deletion mix (common deletion -F (5′-GGC TCA GGC GTT TGT GTA TGAT-3′), common deletion-R (5′-TAT TAA ACA CAA ACT ACC ACC TAC C-3′), and common deletion -P (5′-FAM/ACC ATT GGC/ZEN/AGC CTA G/IBFQ-3′)), 1 μL of 20× Primer/HEX-ZEN RPP30 mix (RPP30-F (5′-GAT TTG GAC CTG CGA GCG-3′), RPP30-R (5′-GCG GCT GTC TCC ACA AGT-3′), and RPP30-P (5′-HEX/CT GAC CTG A/ZEN/A GGC TCT/IBFQ-3′)), and 4 μL of molecular-grade water for the multiplex assay or 1 μL of 20× Primer/FAM ND2 mix (ND2-F (5′-CTT CTG TGG AAC GAG GGT TTA T-3′), ND2-R (5′-CCC GTC ATC TAC TCT ACC ATC T-3′), and ND2-P (5′-FAM/ACA CTC ATC/ZEN/ACA GCG CTA AGC TCG/IBFQ-3′)), and 5 μL of molecular-grade water for the single-plex assay, for a total of 20 μL reaction. As a result, amplification will only occur in the presence of the deletion.

Droplet generation was performed using the Bio-Rad QX200 ddPCR droplet reader according to manufacturer’s protocol. Each reaction was cycled at (i) an initial activation of 95 °C for 10 min, (ii) then 55 cycles of 94 °C for 30 s and 60 °C for 1 min with a ramp speed of 2 °C per second, followed by a final inactivation at 98 °C for 10 min and a 4 °C hold. Primer–probe copies were quantified using the Bio-Rad QX200 ddPCR droplet reader.

### 4.4. Immunoblot Analysis

The DRG and SuN samples from human donors were homogenized and fractionated using a buffer to separate the membrane and cytosolic fractions. Tissues (0.1 g) were homogenized in 0.7 mL of fractionation buffer containing phosphatase and protease inhibitor cocktails. After the determination of the protein content of all samples by the BCA Protein assay (Thermo Fisher Scientific, Rockford, IL, USA), homogenates were loaded (20 mg total protein/lane), separated on 4–12% Bis-Tris gels, electrophoresed in 5% HEPES running buffer, and blotted onto Immobilon-P 0.45 mm membranes using NuPage transfer buffer from Thermo Fisher Scientific. The membranes were blocked and then incubated overnight at 4 °C with primary antibodies. The antibody sets are identified by the following names and dilutions: MFN1: Santa Cruz Biotechnology, Dallas, TX, USA: cat# sc-166644; 1:1000, MFN2: Santa Cruz Biotechnology: cat# sc-515647; 1:1000, DRP1: Santa Cruz Biotechnology; cat# sc-271583; 1:1000, p-DRP1: Cell Signaling, Danvers, MA, USA; cat #3455; 1:1000, and Mito-ETC cocktail: ThermoFisher Catalog # 45-8199; 1:500. All blots were then washed in PBST and then incubated with species-specific IgG conjugated to HRP (American Qualex, San Clemente, CA, USA, cat. no. A102P5) diluted 1:5000 in PBST and visualized with SuperSignal West Femto Maximum Sensitivity Substrate (ThermoFisher Scientific, cat. no. 34096). Images were obtained, and semi-quantitative analysis was performed with the ChemiDoc gel imaging system and Quantity One software Image Lab 6.1 (Bio-Rad, Hercules, CA, USA).

### 4.5. Statistical Analysis

For statistical analysis, the number of mitochondria per cell was averaged for each specimen. The Shapiro–Wilk test was performed to identify the pattern of data distribution. Data are expressed as mean ± SD for normally distributed variables and median (Q1–Q3) for non-normally distributed variables.

For statistical analysis, the Mann–Whitney test was used to compare group data. *p* < 0.05 was considered statistically significant. All statistical analyses and graph illustrations were performed using SPSS version 28 (IBM, Armonk, NY, USA) and GraphPad Prism 10.2 software, LLC, Boston, MA, USA.

## 5. Conclusions

In conclusion, our investigation into mtDNA in the postmortem peripheral nerve tissues of PWH found a reduction in mtDNA copy number in both the DRG and SuN of individuals with DSP, suggesting a potential bidirectional relationship between mitochondrial dysfunction and sensory neuropathy, and indicating that impaired mitochondrial function may contribute to, or result from, neurodegenerative processes in the context of DSP. The increased mtDNA RA carrying the common deletion mutation in the SuN compared to the DRG, particularly in individuals with DSP, highlights the cumulative effect of mtDNA alterations along the course of long nerves. This finding underscores the potential role of mitochondrial genome damage in the development and progression of DSP, with a specific impact on distal nerves. The regional disparity in the expression of mitochondrial ETC complex proteins further emphasizes the complexity of mitochondrial involvement in the pathophysiology of DSP, suggesting a compensatory mechanism within the DRG in response to mitochondrial dysfunction.

The clinical implications of our study suggest that preserving mitochondrial integrity may represent a potential therapeutic target for managing DSP. Strategies aimed at protecting mitochondrial function and preventing mtDNA damage could be explored to mitigate the development and progression of neuropathic symptoms. Furthermore, understanding regional differences in ETC protein expression provides insights into challenges associated with mitochondrial transport within neurons, highlighting potential therapeutic approaches to address specific neuropathic symptoms in DSP.

Exploring the functional consequences of molecular alterations and investigating the direct impact of specific antiretroviral drugs on mitochondrial function could provide a more comprehensive understanding of their underlying mechanisms. Ultimately, our study contributes to the growing body of knowledge aiming to unravel the intricate relationship between HIV, mitochondrial dynamics, and the development of sensory neuropathy, paving the way for potential targeted therapeutic interventions in the future.

## Figures and Tables

**Figure 1 ijms-25-04732-f001:**
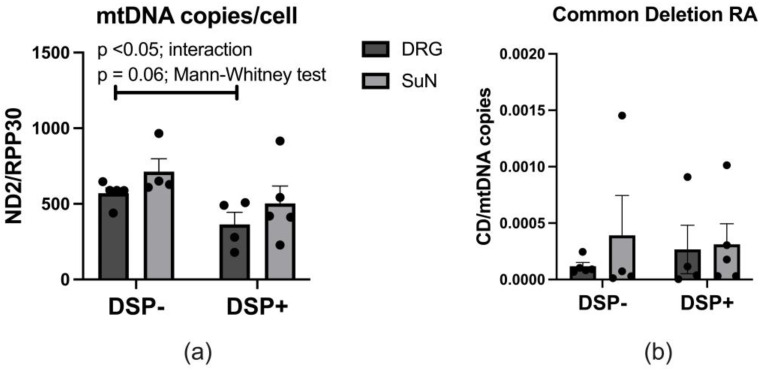
mtDNA copy number and common deletion RA (relative abundance) per DRG (dorsal root ganglion) and SuN (sural nerve) cell. (**a**) ND2/RPP30 gene determined by real-time PCR. (**b**) Quantification of common deletion RA in the DRGs and SuNs of PWH (people with HIV) with and without DSP (distal sensory polyneuropathy) determined by real-time PCR. Mann–Whitney test was used to compare group data. Mean ± SEM. ND2 = NADH de-307 hydrogenase 2. RPP30 = ribonuclease P protein subunit p30.

**Figure 2 ijms-25-04732-f002:**
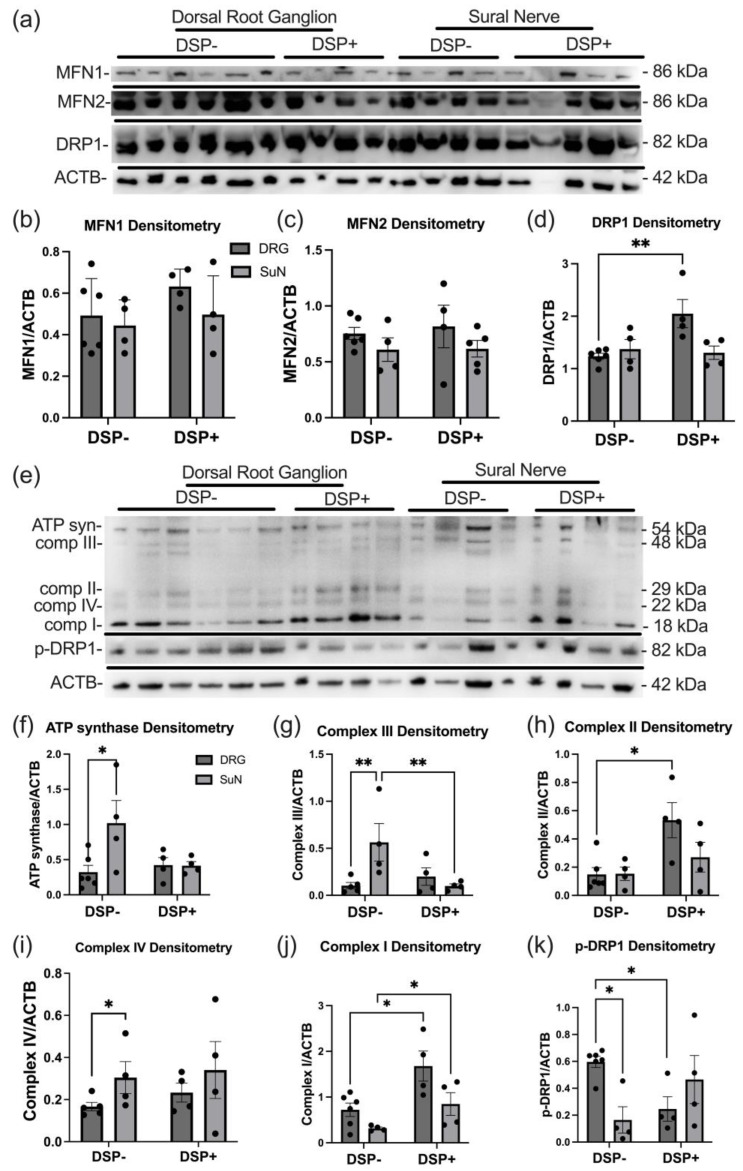
Protein expression levels of ETC (electron transport chain) complex proteins, MFN1 (mitofusion 1), MFN2, and DRP1 (dynamin-1-like protein). (**a**) Immunoblot for mitochondrial dynamic proteins (MFN1, MFN2, and DRP1) and ACTB ((β)-actin) in DRG (dorsal root ganglion) and SuN (sural nerve) lysates of PWH (people with HIV) with and without DSP (distal sensory polyneuropathy). (**b**–**d**) MFN1, MFN2, and DRP1 densitometry normalized to ACTB densitometry. (**e**) Immunoblot for ETC proteins (ATP synthase, Complex III, Complex II, Complex IV, Complex I), p-DRP1, and ACTB in DRG and SuN lysates of PWH DSP− and DSP+. (**f**–**k**) ETC proteins and p-DRP1 densitometry normalized to ACTB densitometry. Mann–Whitney test was used to compare group data. * *p* < 0.05, ** *p* < 0.01. Mean ± SEM.

**Figure 3 ijms-25-04732-f003:**
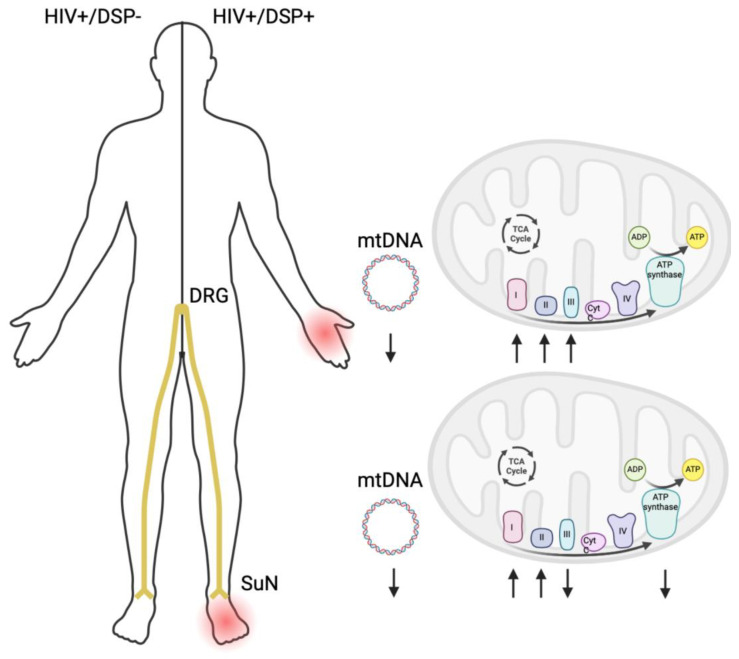
Schematic of alterations in mitochondrial DNA (mtDNA) and electron transport chain (ETC) proteins among people with HIV (PWH) with and without distal sensory polyneuropathy (DSP+/−). In PWH with DSP, mtDNA levels show a decrease in both the dorsal root ganglia (DRG) and the sural nerve (SuN). Additionally, in the DRGs of PWH with DSP, there is an increase observed in Complexes I, II, and III, whereas in the SuNs of the same group, there are increases in levels of Complex I and II and a decrease in levels of Complex III. Furthermore, ATP synthase levels show a decrease in the SuNs of PWH with DSP.

**Table 1 ijms-25-04732-t001:** Clinical characteristics of postmortem samples. DRG and SuN samples from six PWH with no signs of neuropathy and five with two or more signs of neuropathy were used in this study. The participants were predominantly male. On average, PWH with DSP were six years older, and their nadir CD4 levels were reduced by 65% compared to PWH with no signs of neuropathy.

Sensory Peripheral Neuropathy	Gender (M/F)	Age	ART (Current/Past)	D-Drugs * (Current/Past)	Highly Active ART (Yes/No)	Nadir CD4	PMI
No	4/2	42 ± 10.1	4/2	1/4	5/1	14 (2–23)	17.2 ± 15.2
Yes	4/1	48 ± 12.3	4/1	2/3	3/2	8 (7–11)	24 ± 4

* Dideoxynucleoside analog reverse-transcriptase inhibitors (nRTIs) including stavudine and didanosine. ART: antiretroviral therapy. PMI = postmortem interval.

## Data Availability

The datasets generated and/or analyzed during the current study are available in the HIV Neurobehavioral Research Center (HNRC) repository upon reasonable request.

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
