# Peer review of "Mitochondrial DNA and Electron Transport Chain Protein Levels Are Altered in Peripheral Nerve Tissues from Donors with HIV Sensory Neuropathy: A Pilot Study"

_ijms, 2024, doi:10.3390/ijms25094732_

Round 1
Reviewer 1 Report (New Reviewer)
Comments and Suggestions for Authors
The manuscript presents an in-depth investigation into mitochondrial DNA (mtDNA) dynamics and expression of electron transport chain (ETC) proteins in peripheral nerve tissues of people living with HIV (PWH) with distal sensory polyneuropathy (DSP). The findings of the study are highly significant, shedding light on the potential involvement of mitochondrial dysfunction in the development and progression of DSP among PWH. The identification of reduced mtDNA copy number and altered expression of ETC proteins suggests a crucial role for mitochondrial impairment in the pathogenesis of sensory neuropathy, providing valuable insights for further research and potential therapeutic interventions. The manuscript is well-structured and meticulously presents the methods, results, and discussion sections. Overall, the manuscript demonstrates high merit due to its originality, significance, quality of presentation, scientific soundness, and potential interest to readers.
The authors, as it seems to me, have substantially revised the manuscript and made significant corrections. However, I have a few minor comments that should be addressed before the manuscript is published.
Minor comments:
There are a lot of abbreviations in the abstract and in the text, it is difficult to read. The abbreviation common deletion (CD) is generally inappropriate, since it is already used in the scientific literature as circular dichroism. This combination is also difficult to read as: “mtDNA CD RA”.
Lines 86-87. The phrase "addressing broader various metabolic and neurodegenerative disorders associated 37" seems incomplete.
Line 117. "There was a non-significant increase. There was a non-significant increase...", the repetition of "There was a non-significant increase " please remove for clarity.
Line 156. "MFN 1/2". Please write as "MFN1 and MFN2" for clarity.
Line 157. "Mean+SEM". Please expand as "mean ± SEM" for clarity.
Line 188. “d-drugs." It's unclear what "d-drugs" refers to.
Line 296. "Fischer". Please correct to "Fisher".
The authors were correct in supplementing the main text of the manuscript with additional materials. However, the original materials of blots in the supplementary materials of the article should include explanations, as well as labeling of samples. Readers should understand what is depicted in these photographs even without referring to the main text of the article.
Author Response
Response to Reviewer 1
- There are a lot of abbreviations in the abstract and in the text, it is difficult to read. The abbreviation common deletion (CD) is generally inappropriate, since it is already used in the scientific literature as circular dichroism. This combination is also difficult to read as: “mtDNA CD RA”.
Author response: We appreciate the reviewer’s comment and attention to detail. We have changed the abbreviation “CD” to “common deletion” throughout the manuscript.
- Lines 86-87. The phrase "addressing broader various metabolic and neurodegenerative disorders associated 37" seems incomplete.
Author response: We thank the reviewer for bringing this to our attention. Please see Page 2, lines 85-88. We have revised this sentence as follows:
“Further elucidating the molecular pathways involved in HIV-induced mitochondrial dysfunction may unveil novel targets for intervention, not only in the context of HIV treatment but also in addressing broader mitochondrial-related diseases.”
- Line 117. "There was a non-significant increase. There was a non-significant increase...", the repetition of "There was a non-significant increase " please remove for clarity.
Author response: We have removed the repetitive phrasing on Page 3, lines 119-120 as follows:
“There was a non-significant increase in mtDNA copy number per cell in SuN compared to DRG in both DSP- and DSP+ decedents (Figure 1a)”
- Line 156. "MFN 1/2". Please write as "MFN1 and MFN2" for clarity.
Author response: Please see Page 5, lines 163-164 which now read as follows:
“Protein expression levels of ETC (electron transport chain) complex proteins, MFN1 (mitofusion 1), MFN2, and DRP1 (Dynamin-1-like protein).”
- Line 157. "Mean+SEM". Please expand as "mean ± SEM" for clarity.
Author response: We have made this change on Page 5, lines 170-171 as follows:
“All data were analyzed with 2-way ANOVA with post-hoc Tukey’s. *p < 0.05, **p < 0.01. Mean ± SEM.”
- Line 188. “d-drugs." It's unclear what "d-drugs" refers to.
Author response: We appreciate the reviewer’s comment. We have now defined d-drugs and added the specific names of d-drugs on Page 6, lines 185-186 as follows:
“certain d-drugs (dideoxynucleoside analog reverse transcriptase inhibitors), specifically Stavudine, Didanosine, and Zalcitabine”
- Line 296. "Fischer". Please correct to "Fisher".
Author response: We appreciate the reviewer’s attention to detail. We have corrected the name from "Fischer" to "Fisher on Page 9, lines 306-307 as follows:
“Enzymatic digestion – Extracted DNA was enzymatically digested using the BamHI enzyme (ThermoFisher Scientific, New York, USA).”
- The authors were correct in supplementing the main text of the manuscript with additional materials. However, the original materials of blots in the supplementary materials of the article should include explanations, as well as labeling of samples. Readers should understand what is depicted in these photographs even without referring to the main text of the article.
Author response: We have included a pdf document that includes all original blot images with labels for molecular weights and protein bands.
Reviewer 2 Report (New Reviewer)
Comments and Suggestions for Authors
Mitochondrial DNA and electron transport chain protein levels are altered in peripheral nerve tissues from donors with HIV sensory neuropathy: A pilot study
The present study investigates the mitochondrial DNA and electron transport chain protein changes in the dorsal root ganglion and sural nerve from autopsied people with HIV. Even though the population size is very low, the approach and the overall design of the study are good. However, the authors should address the following concerns.
1. It is stated in the ‘Discussion’ section that
“This investigation utilized a combination of molecular and histopathological methods to analyze mtDNA dynamics and ETC protein expression in post-mortem DRG and SuN specimens from individuals with and without DSP.”
However, no information is provided in the article about the histopathological observations. Also, it has been presented as a limitation.
2. Figures should be self-explanatory. Include the details of statistical comparison in the figure legends for better clarity.
3. It is suggested to recheck the quantification of the immunoblots using densitometry. Since the ACTB bands show too much variability, the observed changes and statistical significance need rechecking. Especially for ETC proteins.
4. Abbreviations should be expanded in the main text in addition to the abstract.
5. A schematic diagram should be included showing the overall findings of the study.
Comments on the Quality of English Language
Minor editing of the English language is required.
Author Response
- It is stated in the ‘Discussion’ section that “This investigation utilized a combination of molecular and histopathological methods to analyze mtDNA dynamics and ETC protein expression in post-mortem DRG and SuN specimens from individuals with and without DSP.” However, no information is provided in the article about the histopathological observations. Also, it has been presented as a limitation.
Author response: We have corrected this oversight by removing the term “histopathological methods”. Please see Page 6, lines 175-177. The updated sentence reads as follows:
“This study used ddPCR to analyze mtDNA and immunoblot to assess the expression of mitochondrial dynamics and ETC proteins in post-mortem DRG and SuN specimens from individuals with and without DSP.”
- Figures should be self-explanatory. Include the details of statistical comparison in the figure legends for better clarity.
Author response: We have revised the figure legends as suggested. Please see the below details:
Page 4, lines 132-137:
“Figure 1. Figure 1. mtDNA copy number and common deletion RA (relative abundance) per DRG (dorsal root ganglion) and SuN (sural nerve) cells. (a) ND2/RPP30 gene determined by real-time PCR. (b) Quantification of common deletion RA in the DRG and SuN of PWH (people with HIV) with and without DSP (distal sensory polyneuropathy) determined by real-time PCR. Statistical analysis by 2-way ANOVA and post-hoc Tukey’s. Mean ± SEM. ND2 = NADH de-307 hydrogen-ase 2. RPP30 = ribonuclease P protein subunit p30.”
Page 5, lines 163 –171:
“Figure 2. Protein expression levels of ETC (electron transport chain) complex proteins, MFN1 (mitofusion 1), MFN2, and DRP1 (Dynamin-1-like protein). (a) Immunoblot for mitochondrial dynamics proteins (MFN1, MFN2, and DRP1) and ACTB ((β)-actin) in DRG (dorsal root ganglion) and SuN (sural nerve) lysates of PWH (people with HIV) with and without DSP (distal sensory polyneuropathy). (b-d) MFN1, MFN2, and DRP1 densitometry normalized to ACTB densitometry. (e) Immunoblot for ETC proteins (ATP synthase, complex III, complex II, complex IV, complex I), p-DRP1, and ACTB in DRG and SuN lysates of PWH DSP- and DSP+. (f-k) ETC proteins and p-DRP1 densitometry normalized to ACTB densitometry. All data was analyzed with 2-way ANOVA with post-hoc Tukey’s. *p < 0.05, **p < 0.01. Mean ± SEM.”
Page 3, line 114:
Table 1. PMI has been defined as postmortem interval.”
- It is suggested to recheck the quantification of the immunoblots using densitometry. Since the ACTB bands show too much variability, the observed changes and statistical significance need rechecking. Especially for ETC proteins.
Author response: Thank you for the comment. We have re-checked the quantification and validated the presented data.
- Abbreviations should be expanded in the main text in addition to the abstract.
Author response: We appreciate the reviewer’s comment. Abbreviations have been defined the first time they appear in the main text.
Page 1, line 33: antiretroviral therapies (ART)
Page 2, line 45: mitochondrial DNA (mtDNA)
Page 2, line 93: dorsal root ganglion (DRG)
Page 2, line 75: sural nerve (SuN)
Page 9, line 313: digital droplet polymerase chain reaction (ddPCR)
Page 2, line 45: electron transport chain (ETC)
- A schematic diagram should be included showing the overall findings of the study.
Author response: We agree that a schematic diagram illustrating the overall findings of the study would be beneficial for readers to grasp the key results more effectively. We created a schematic diagram and incorporated it into the manuscript to provide a visual summary of our study's findings.
Figure 3. Schematic of alterations in mitochondrial DNA (mtDNA) and electron transport chain (ETC) proteins among people with HIV (PWH) with and without distal sensory polyneuropathy (DSP+/-). In PWH with DSP, mtDNA levels show a decrease in both the dorsal root ganglia (DRG) and the sural nerve (SuN). Additionally, in the DRG of PWH with DSP, there is an increase observed in complexes I, II, and III, whereas in the SuN of the same group, there is increases in levels of complex I and II and a decrease in levels of complex III. Furthermore, ATP synthase levels show a decrease in the SuN of PWH with DSP.
Round 2
Reviewer 2 Report (New Reviewer)
Comments and Suggestions for Authors
The manuscript can be accepted for publication.
This manuscript is a resubmission of an earlier submission. The following is a list of the peer review reports and author responses from that submission.
Round 1
Reviewer 1 Report
Comments and Suggestions for Authors
Study investigated mtDNA, mt fission and fusion protein, and mt electron transport chain protein changes in the dorsal root ganglion (DRG) and sural nerve (SuN) from 11 autopsied PWH. In antemortem standardized assessments, six had no or one sign of DSP, and five exhibited two or more DSP signs. Digital droplet polymerase chain reaction (ddPCR) measured mtDNA quantity and the common deletion (CD) in isolated DNA. Authors discovered lower mtDNA copy numbers in DSP+ donors. SuN exhibited a higher proportion of mtDNA CD than DRGs in both groups. Mitochondrial electron transport chain (ETC) proteins were altered in DRG of DSP+ compared to DSP- donors, particularly complex I. Conclusion: Authors concluded that mtDNA quantity and increased CD abundance may contribute to DSP in PWH, indicating diminished mitochondrial activity in sensory neurons. Accumulated ETC proteins in the DRG imply impaired mitochondrial transport to the sensory neuron's distal portion. Study is interesting, though limited by number of specimens and absence of specimens derived from control group without HIV and DSP to establish ‘baseline’ for measured parameters. Other limitations are listed below.
1. Histopathologic assessment of these samples is need to assess neuropathologic changes in each sample.
2. There is no information about postmortem interval which may affect degradation of DNA and proteins.
3. ART is mentioned in Materials and Methods, but its role in mitochondrial dysfunction difficult to assess in present form.
Author Response
Response to Reviewer 1
- Histopathologic assessment of these samples is need to assess neuropathologic changes in each sample.
Author response: The authors agree with the reviewer’s astute observation that histopathologic analysis would improve the study, but, unfortunately, fixed specimens were unavailable at the time of analyses, and thus, we plan to conduct such experiments in future studies in a larger cohort. However, the findings we present are novel regarding the specimens used (dorsal root ganglion and sural nerve tissues from people with HIV on ART), while also being consistent with previous studies that have been supportive of a role for mitochondrial dysfunction in DSP in PWH.
We have included the lack of histopathologic assessment as a limitation on Page 7, Lines 231-236: “We aim to complete future studies with a larger number of specimens from a diverse group of decedents. While the analyses of biochemical land molecular of DRG and SuN are novel in the context of DSP in PWH, histopathologic assessments of the specimens are needed to confirm that neuropathological changes are consistent with the protein and mtDNA alterations described here in this study. It is important to include such assessments in future studies.”
- There is no information about postmortem interval which may affect degradation of DNA and proteins.
Author response: The postmortem interval (PMI) is an important characteristic of pathologic studies and must be considered considering the data. While the average PMI is ~17 and ~24 h for the DSP- and DSP+ groups, respectively, the differences are not significant when scrutinized by the student’s T-test. Moreover, the changes in mtDNA and protein levels between the groups are not reflective of increased degradation in the group with the larger PMI. For example, half of the proteins measured by western blot are at higher levels in the group with the higher PMI. That said, the authors agree with the reviewer that the PMI, and discussion when warranted, should be included in the manuscript.
We have included the PMI mean and standard deviation for each group on Page 3, lines 104-106: “The difference in postmortem intervals (PMI) between the DSP- (mean = 17.2 hours, standard deviation = 15.2 hours) and DSP+ (mean = 24 hours, standard deviation = 4 hours) groups were not significant.”
We have included a discussion of the PMI for these groups as a limitation on Page 7, lines 236-240: The PMI for the specimens included in this study are higher than ideal for neuropathological assessments, and effort will be made in the future to examine tissues with PMI under 8 hours. However, the differences in PMI between the groups are not significant, and the changes in mtDNA and protein levels between the groups are not reflective of increased degradation in the group with the larger PMI.
- ART is mentioned in Materials and Methods, but its role in mitochondrial dysfunction difficult to assess in present form.
Author Response: We are thankful for the reviewer’s comment and attention to detail. We appreciate your concern regarding the role of ART in mitochondrial dysfunction. It is important to note that the impact of ART on mitochondrial function has been studied in previous research. Given that all participants in our study were either currently undergoing cART or had a history of previous exposure to antiretroviral medications (please see page 3, line 141-143 and table 1), we did not specifically assess the role of ART in our study. However, we did add a brief discussion to the discussion section indicating that different ART drugs can have different effects. Please see Page 7, lines 243-245:
“The impact of ART on mitochondrial function varies widely depending on the specific drugs used, their dosage, and individual patient factors.”
Instead, our focus was on elucidating mitochondrial dysfunction in the context of HIV infection, particularly in relation to DSP. We have also discussed the role of ART in DSP in the introduction section of our paper. We hope this clarification addresses your concern. Please see Pages 1-3, lines 44-102.”
“ART induces mitochondrial toxicity, impairs oxidative phosphorylation, increases reactive oxygen species, reduces ATP synthesis, and alters mitochondrial biogenesis (17), effects which have been demonstrated to occur in sensory neurons and have been implicated in antiretroviral toxic neuropathy (ATN) and peripheral neuropathy(17-19). Further, HIV and ART have been implicated in the inhibition of the gamma DNA polymerase, the enzyme responsible for mtDNA replication (20), compromising the mitochondrial genome and leading to mitochondrial dysfunction (21).”
Reviewer 2 Report
Comments and Suggestions for Authors
This manuscript topic is of interest and adds data to the theory of mitochondrial dysfunctions as a cause of neuropathies among people living with HIV. Overall, the quality of the manuscript is low and does not meet the standard of IJMS. Listed below some of my major concerns.
Introduction
- An update of the literature is definitively needed. The same objective was addressed by Roda et al. in 2021 partly using some of the same biopsies (n=67).
- Line 77. The notion of heteroplasmy is required at this stage to better describe the association of neuroplathy and mitochondrial dysfunction.
- Line 87-89. The hypothesis proposed by the authors is not evidence based, associating symptoms and mt dysfunction is not straightforward. Did the authors also hypothesized that a difference between the two specimens was expected? It is unclear for me.
- Can the authors clearly state their objectives at the end of the introduction?
Method :
- The first § is not necessary. The overall strategy to meet the objectives can be included in the introduction is authors feel it is needed.
- Line 286. The number of the IRB approval is usually mentioned.
- Table 1 is a result, not a method
- Line 297. Can the authors add references to the criteria/scale they have used?
- Line 311. Description of the ddPCR is too minimal. It is not possible to repeat these experiments with the given information. A complete list of primer, a reference, and a determination of the lower limit of detection would be very helpful.
- Line 323. No list of antibodies, thus impossible to reproduce. The revelation step is not described.
- Line 331. The description of the statistical method is too short. I don’t get the real meaning of “the number of mitochondria per compartment was averaged for each specimen”. Does compartment means cells?
- Were the test adjusted for low number of samples?
Result
- Table 1 should appear at line 119.
- Can the different variables described at the end of this § be listed in the Table 1?
- Line 121. In this section, give results only and no rational or interpretation of the results.
- If sensitive analysis, please give a justification?
- Figure 3A refers to t-test whereas the method section refers to KW test. Please correct the error?
- For the common deletion, it is impossible to evaluate the data without the definition of their LLOD, given the low percentage.
Discussion
- Line 185-188. This information is better placed in the introduction
- Line 197-200. The bottom line is that the level of CD RA mtDNA is low or very low. How can this level impact on the function of these nerves, given that heteroplasmy may compensate these deficiencies.
- Regarding the variation of the expression of the ETC proteins, can the authors discuss whether a postmortem sampling could explain, or partly explain these variations?
- Line 243. The limitation section rather suggests future studies.
- Line 335. Please reduce this section to the main message.
Minor points :
Figure 1 and 2 are not contributive to the understanding of the manuscript. It rather fit in a PhD dissertation.
Line 148. First time use of DRP1 and ACTB
Line 312 First time use of RA
Author Response
Introduction
- An update of the literature is definitively needed. The same objective was addressed by Roda et al. in 2021 partly using some of the same biopsies (n=67).
Response. We thank reviewer 2 for this insightful comment. We have added information regarding the suggested study with the following text on:
Page 1, lines 40-51: Neurons depend on ATP for axonal transport and maintaining ionic gradients, which is crucial for generating action potentials and facilitating synaptic activity (10). Both ART and HIV proteins (ex: gp120) have been implicated in HIV-DSP (9, 11), likely through mechanisms that involve mitochondrial function (12-14). HIV infection impairs mitochondrial respiration, ETC activity, and mtDNA replication (15, 16). ART induces mitochondrial toxicity, impairs oxidative phosphorylation, increases reactive oxygen species, reduces ATP synthesis, and alters mitochondrial biogenesis (17), effects which have been demonstrated to occur in sensory neurons and have been implicated in antiretroviral toxic neuropathy (ATN) and peripheral neuropathy(17-19). Further, HIV and ART have been implicated in the inhibition of the gamma DNA polymerase, the enzyme responsible for mtDNA replication (20), compromising the mitochondrial genome and leading to mitochondrial dysfunction (21).
Page 2, lines 63-76: Therefore, cells with high levels of the CD mutation should exhibit more pronounced mitochondrial dysfunction compared to cells with lower levels of this mutation (27-29). In a noteworthy prior study of 67 HIV-positive individuals on ART, there was an inverse correlation between mtDNA deletions and peripheral neuropathy as assessed by intraepidermal nerve fiber density and sural nerve amplitude, further suggesting a potential role for mitochondrial dysfunction in the neuropathic processes associated with HIV and its treatment (30, 31).
Axonal mitochondria are assembled in the neuronal cell body and transported down the length of axons utilizing microtubule-based machinery, eventually becoming an-chored in the axon according to energy demands (32). Therefore, mitochondria located in the distal axons of long peripheral nerves, such as the SuN, are at increased risk of accumulating mtDNA mutations (16, 30, 33) and are heavily reliant on quality control measures, including fission and fusion (34). Processes that can be impaired by neuroinflammatory processes such as HIV (35, 36).
- Line 77. The notion of heteroplasmy is required at this stage to better describe the association of neuropathy and mitochondrial dysfunction.
We thank reviewer 2 for this valuable suggestion. We have addressed heteroplasmy in the introduction section with the following text on Page 2, lines 58-65:
“CD mutations are somatic mutations. Within cells, CD mutations exist in heteroplasmy along with wild-type mitochondria. The proportion of mutated mtDNA relative to normal mtDNA can vary, leading to varying degrees of mitochondrial dysfunction depending on this ratio. Wild-type mitochondria are able to compensate for biochemical deficits attributable to CD mitochondria up to a certain threshold, beyond which pathologic phenotype emerges. Therefore, cells with high levels of the CD mutation should exhibit more pronounced mitochondrial dysfunction compared to cells with lower levels of this mutation (27-29).”
- Line 87-89. The hypothesis proposed by the authors is not evidence based, associating symptoms and mt dysfunction is not straightforward. Did the authors also hypothesized that a difference between the two specimens was expected? It is unclear for me.
Response: We thank reviewer 2 for this insightful comment. We expected to see differences between both DSP+ and DSP- groups as well as between DRG and SuN samples. We have clarified our hypothesis in the introduction section with the following text on Page 2, lines 91-94:
“We hypothesized that we would see increased mitochondrial mutations and decreased fission, fusion, and ETC protein concentrations in DSP+ individuals compared to DSP- individuals and that these effects would be more pronounced in SuN.”
Additionally, we have expanded the introduction to include additional background information to help clarify the association between mitochondrial dysfunction, HIV, ART, and peripheral distal neuropathy. Please see Page 1, lines 44-51:
“ART induces mitochondrial toxicity, impairs oxidative phosphorylation, increases reactive oxygen species, reduces ATP synthesis, and alters mitochondrial biogenesis (17), effects which have been demonstrated to occur in sensory neurons and have been implicated in antiretroviral toxic neuropathy (ATN) and peripheral neuropathy(17-19). Further, HIV and ART have been implicated in the inhibition of the gamma DNA polymerase, the enzyme responsible for mtDNA replication (20), compromising the mitochondrial genome and leading to mitochondrial dysfunction (21).”
As well as Page 2, lines 65-76:
“In a noteworthy prior study of 67 HIV-positive individuals on ART, there was an inverse correlation between mtDNA deletions and peripheral neuropathy as assessed by intraepidermal nerve fiber density and sural nerve amplitude, further suggesting a potential role for mitochondrial dysfunction in the neuropathic processes associated with HIV and its treatment (30, 31).
Axonal mitochondria are assembled in the neuronal cell body and transported down the length of axons utilizing microtubule-based machinery, eventually becoming an-chored in the axon according to energy demands (32). Therefore, mitochondria located in the distal axons of long peripheral nerves, such as the SuN, are at increased risk of accumulating mtDNA mutations (16, 30, 33) and are heavily reliant on quality control measures, including fission and fusion (34). Processes that can be impaired by neuroinflammatory processes such as HIV (35, 36).”
- Can the authors clearly state their objectives at the end of the introduction?
Response: We thank reviewer 2 for this insightful suggestion. We have clarified our objectives in the introduction section with the following text on Page 2, line 88-91:
“the overall aim of this study was to compare the differences in mitochondrial mutations (CD number), nuclear-encoded fission and fusion proteins (MFN1, MFN2, and DRP1) and mtDNA-encoded ETC proteins (ATP-synthase, complex I, II, III, IV) in PWH with DSP (DSP+) and without DSP (DSP- ) and between DRG and SuN samples.”
Method
- The first § is not necessary. The overall strategy to meet the objectives can be included in the introduction is authors feel it is needed.
Response: We thank reviewer 2 for this valuable suggestion. We have deleted this paragraph. We have moved the sentence pertaining to the DRG and SuN specimens to 4.1 study population on Page 8, lines 274-275 as follows:
“DRG and SuN specimens were obtained from this cohort posthumously and processed as below.”
- Line 286. The number of the IRB approval is usually mentioned.
Response: We thank reviewer 2 for this valuable suggestion. We added the IRB approval code #080323 to section 4.1. Please see Page 8, lines 271-273.
“An IRB approved this research with the approval code #080323, and each participant gave informed consent.”
- Table 1 is a result, not a method
Response: We thank reviewer 2 for this observation. We have moved Table 1 to the results section on Page 3, lines 107-112:
Table 1. Clinical characteristics of postmortem samples. DRG and the SuN from six PWH with no signs of neuropathy and 5 with two or more signs of neuropathy were used in this study. The participants were predominantly male. On average, PWH with DSP were six years older, and their nadir CD4 levels were reduced by 65 % compared to PWH with no signs of neuropathy.
|
Sensory Peripheral Neuropathy |
Gender (M/F) |
Age |
ART (current/past) |
D-drugs* (current/past) |
Highly active ART (yes/no) |
Nadir CD4 |
PMI |
|
No |
4/2 |
42 ± 10.1 |
4/2 |
1/4 |
5/1 |
14 (2-23) |
17.2 ± 15.2 |
|
Yes |
4/1 |
48 ± 12.3 |
4/1 |
2/3 |
3/2 |
8 (7-11) |
24 ± 4 |
* Dideoxynucleoside analog reverse transcriptase inhibitors (nRTIs) including stavudine or Didanosine. ART: Anti-retroviral therapy.
- Line 297. Can the authors add references to the criteria/scale they have used?
Response: We thank reviewer 2 for this valuable suggestion. We added references for and clarification of the criteria used to diagnosis HIV-associated DSP to the methods section 4.2 with the following text on Page 8, lines 278-279:
“HIV-associated DSP was determined using the CHARTER study protocol which has previously been described in detail (PMID: 33565583). Briefly”
And on Page 8, lines 289-290:
“The presence of at least 2 signs bilaterally was considered to be evidence of DSP (46).”
- Line 311. Description of the ddPCR is too minimal. It is not possible to repeat these experiments with the given information. A complete list of primer, a reference, and a determination of the lower limit of detection would be very helpful.
Response: We thank reviewer 2 for this valuable suggestion. We have expanded the 4.3 methods section to include a more comprehensive description of the methodology, a comprehensive list of primers and references for the assay methodology with the following text on Page 8, lines 294-331:
“DNA quantification was performed using the highly sensitive droplet digital PCR platform (42). Genomic DNA was extracted from DRG and SuN specimens and then fragmented using three different methods:
Enzymatic digestion - Extracted DNA was enzymatically digested using BamHI enzyme (Thermo Fischer Scientific, New York, USA). 250 ng DNA in 10 μL 10 mM Tris-EDTA (TE) buffer was added to 10 μL 0.2× BamHI Buffer containing 10 U BamHI enzyme and incubated at 37 °C for 1 h.
Sonication - Sonication was performed by adding 375 ng DNA in 30 μL TE buffer to a microtube. A range of sonication target lengths of 200 bp, 500 bp, 800 bp, 2 kb, and 5 kb were tested using a Covaris M220 Focused Ultrasonicator. Quantification of target sequences using ddPCR was used to determine which sonication length was optimal.
QIAshredder spin columns – 375 ng DNA in 30 μL TE buffer was introduced into the QIAshredder spin column and centrifuged for 2 min at 13,000 rpm.
The resulting fragmented DNA concentration was 12.5 ng/μL regardless of the fragmentation method used. Fragmented DNA was then introduced into the ddPCR reaction.
The mtDNA copy number was measured by targeting the mitochondrial ND2, while the mtDNA CD was measured using a primer-probe set targeting the ends of the deletion. RPP30 was used as a cellular control as 2 copies are present in each cell. CD and RPP30 assays were multiplexed using 50 ng of DNA per replicate, while the ND2 assay was performed alone using 50 pg of DNA per replicate. Both assays were run in triplicate (43). Quantification was performed as follows: 50 ng or 50 pg of DNA (in 4 μL) were added to a master mix consisting of 10 μL of 2× Bio-Rad supermix for probes, 1 μL of 20× Primer/FAM-ZEN CD mix (CD-F (5′-GGC TCA GGC GTT TGT GTA TGAT-3′), CD-R (5′-TAT TAA ACA CAA ACT ACC ACC TAC C-3′), and CD-P (5′-FAM/ACC ATT GGC/ZEN/AGC CTA G/IBFQ-3′)), 1 μL of 20× Primer/HEX-ZEN RPP30 mix (RPP30-F (5′-GAT TTG GAC CTG CGA GCG-3′), RPP30-R (5′-GCG GCT GTC TCC ACA AGT-3′), and RPP30-P (5′-HEX/CT GAC CTG A/ZEN/A GGC TCT/IBFQ-3′)), and 4 μL of molecular grade water for the multiplex assay or 1 μL of 20× Primer/FAM ND2 mix (ND2-F (5′-CTT CTG TGG AAC GAG GGT TTA T-3′), ND2-R (5′-CCC GTC ATC TAC TCT ACC ATC T-3′), and ND2-P (5′-FAM/ACA CTC ATC/ZEN/ACA GCG CTA AGC TCG/IBFQ-3′)), and 5 μL of molecular grade water for the singleplex assay, for a total of 20 μL reaction. As a result, amplification will only occur in the presence of the deletion.
Droplet generation was performed using the Bio-Rad QX200 ddPCR droplet reader according to manufacturer protocol. Each reaction was cycled at (i) initial activation of 95 °C for 10 min, (ii) then 55 cycles of 94 °C for 30 s and 60 °C for 1 min with a ramp speed of 2 °C per second, followed by a final inactivation at 98 °C for 10 min and a 4 °C hold. Primer-probe copies were quantified using the Bio-Rad QX200 ddPCR droplet reader.”
- Line 323. No list of antibodies, thus impossible to reproduce. The revelation step is not described.
Response: We thank reviewer 2 for this valuable suggestion. We have expanded the 4.4 methods section to include a more comprehensive description of revelation methodology, including a list of antibodies, with the following text on Page 9, lines 340-349:
“The antibody sets are identified by the following names and dilution: MFN1: Santa Cruz Biotechnology: cat# sc-166644; 1:1000, MFN2: Santa Cruz Biotechnology: cat# sc-515647; 1:1000, DRP1: Santa Cruz Biotechnology; cat# sc-271583; 1:1000, p-DRP1: Cell Signaling; cat #3455; 1:1000, and Mito-ETC cocktail: ThermoFisher Catalog # 45-8199; 1:500. All blots were then washed in PBST, and then incubated with species-specific IgG conjugated to HRP (American Qualex, cat. no. A102P5) diluted 1:5,000 in PBST and visualized with SuperSignal West Femto Maximum Sensitivity Substrate (ThermoFisher Scientific, cat. no. 34096). Images were obtained, and semi-quantitative analysis was performed with the ChemiDoc gel imaging system and Quantity One software (Bio-Rad).”
- Line 331. The description of the statistical method is too short. I don’t get the real meaning of “the number of mitochondria per compartment was averaged for each specimen”. Does compartment means cells?
Response: We thank reviewer 2 for this valuable suggestion. We have added additional information regarding the statistical measures and have changed our wording from compartment to cells with the following text on Page 9, lines 351-358:
“For statistical analysis, the number of mitochondria per cell was averaged for each specimen. The Shapiro-Wilk test was performed to identify the pattern of data distribution. Data were expressed as mean ± SD for normally distributed variables and median (Q1–Q3) for non-normally distributed variables.
For statistical analysis, the Mann-Whitney and Kruskal-Wallis tests with Dunn’s post-test were used to compare group data. P<0.05 was considered statistically significant. All statistical analyses and graph illustrations were performed using SPSS version 28 (IBM, USA) and GraphPad Prism 10.2 software, LLC.”
- Were the test adjusted for low number of samples?
Thank you for your comment. Since the number of samples was relatively low, we conducted the Shapiro-Wilk test to identify the pattern of data distribution and ran the Mann-Whitney test with Dunn’s post-test for group comparisons accordingly.
Result
- Table 1 should appear at line 119.
Response: We thank reviewer 2 for this valuable observation. We have moved Table 1 to the Results section. Please see Page 3, lines 107-112:
- Can the different variables described at the end of this § be listed in the Table 1?
Response: We thank reviewer 2 for this valuable observation. We have revised Table 1 to include the variables reported in the description. Please see Page 3, lines 107-112.
- Line 121. In this section, give results only and no rational or interpretation of the results.
Response: We thank reviewer 2 for this valuable suggestion. We have removed the interpretation from the results section.
- If sensitive analysis, please give a justification?
Response: We thank reviewer 2 for this comment. We justified the statistical tests in the methods section with the following text on Page 9, lines 351-358:
“For statistical analysis, the number of mitochondria per cell was averaged for each specimen. The Shapiro-Wilk test was performed to identify the pattern of data distribution. Data were expressed as mean ± SD for normally distributed variables and median (Q1–Q3) for non-normally distributed variables. For statistical analysis, the Mann-Whitney and Kruskal-Wallis tests with Dunn’s post-test were used to compare group data. P<0.05 was considered statistically significant. All statistical analyses and graph illustrations were performed using SPSS version 28 (IBM, USA) and GraphPad Prism 10.2 software, LLC.”
- Figure 3A refers to t-test whereas the method section refers to KW test. Please correct the error?
Response: We thank reviewer 2 for this valuable observation. We have clarified the statistical tests in Figures 1 and 2 and in the methods section with the following text on Page 9, lines 351-358:
“For statistical analysis, the number of mitochondria per cell was averaged for each specimen. The Shapiro-Wilk test was performed to identify the pattern of data distribution. Data were expressed as mean ± SD for normally distributed variables and median (Q1–Q3) for non-normally distributed variables.
For statistical analysis, the Mann-Whitney and Kruskal-Wallis tests with Dunn’s post-test were used to compare group data. P<0.05 was considered statistically significant. All statistical analyses and graph illustrations were performed using SPSS version 28 (IBM, USA) and GraphPad Prism 10.2 software, LLC.”
- For the common deletion, it is impossible to evaluate the data without the definition of their LLOD, given the low percentage.
Response: We thank reviewer 2 for this insightful critique. We have expanded the 4.3 methods section to include a more comprehensive description of the methodology, a comprehensive list of primers and references for the assay methodology with the following text on Page 8, lines 294-331:
“DNA quantification was performed using the highly sensitive droplet digital PCR platform (42). Genomic DNA was extracted from DRG and SuN specimens and then fragmented using three different methods:
Enzymatic digestion - Extracted DNA was enzymatically digested using BamHI enzyme (Thermo Fischer Scientific, New York, USA). 250 ng DNA in 10 μL 10 mM Tris-EDTA (TE) buffer was added to 10 μL 0.2× BamHI Buffer containing 10 U BamHI enzyme and incubated at 37 °C for 1 h.
Sonication - Sonication was performed by adding 375 ng DNA in 30 μL TE buffer to a microtube. A range of sonication target lengths of 200 bp, 500 bp, 800 bp, 2 kb, and 5 kb were tested using a Covaris M220 Focused Ultrasonicator. Quantification of target sequences using ddPCR was used to determine which sonication length was optimal.
QIAshredder spin columns – 375 ng DNA in 30 μL TE buffer was introduced into the QIAshredder spin column and centrifuged for 2 min at 13,000 rpm.
The resulting fragmented DNA concentration was 12.5 ng/μL regardless of the fragmentation method used. Fragmented DNA was then introduced into the ddPCR reaction.
The mtDNA copy number was measured by targeting the mitochondrial ND2, while the mtDNA CD was measured using a primer-probe set targeting the ends of the deletion. RPP30 was used as a cellular control as 2 copies are present in each cell. CD and RPP30 assays were multiplexed using 50 ng of DNA per replicate, while the ND2 assay was performed alone using 50 pg of DNA per replicate. Both assays were run in triplicate (43). Quantification was performed as follows: 50 ng or 50 pg of DNA (in 4 μL) were added to a master mix consisting of 10 μL of 2× Bio-Rad supermix for probes, 1 μL of 20× Primer/FAM-ZEN CD mix (CD-F (5′-GGC TCA GGC GTT TGT GTA TGAT-3′), CD-R (5′-TAT TAA ACA CAA ACT ACC ACC TAC C-3′), and CD-P (5′-FAM/ACC ATT GGC/ZEN/AGC CTA G/IBFQ-3′)), 1 μL of 20× Primer/HEX-ZEN RPP30 mix (RPP30-F (5′-GAT TTG GAC CTG CGA GCG-3′), RPP30-R (5′-GCG GCT GTC TCC ACA AGT-3′), and RPP30-P (5′-HEX/CT GAC CTG A/ZEN/A GGC TCT/IBFQ-3′)), and 4 μL of molecular grade water for the multiplex assay or 1 μL of 20× Primer/FAM ND2 mix (ND2-F (5′-CTT CTG TGG AAC GAG GGT TTA T-3′), ND2-R (5′-CCC GTC ATC TAC TCT ACC ATC T-3′), and ND2-P (5′-FAM/ACA CTC ATC/ZEN/ACA GCG CTA AGC TCG/IBFQ-3′)), and 5 μL of molecular grade water for the singleplex assay, for a total of 20 μL reaction. As a result, amplification will only occur in the presence of the deletion.
Droplet generation was performed using the Bio-Rad QX200 ddPCR droplet reader according to manufacturer protocol. Each reaction was cycled at (i) initial activation of 95 °C for 10 min, (ii) then 55 cycles of 94 °C for 30 s and 60 °C for 1 min with a ramp speed of 2 °C per second, followed by a final inactivation at 98 °C for 10 min and a 4 °C hold. Primer-probe copies were quantified using the Bio-Rad QX200 ddPCR droplet reader.”
Discussion
- Line 185-188. This information is better placed in the introduction
Response: We appreciate the reviewer’s comment. We made some changes in those lines so they fit a suitable place in the introduction section on Page 2, lines 65-69:
In a noteworthy prior study of 67 HIV-positive individuals on ART, there was an inverse correlation between mtDNA deletions and peripheral neuropathy as assessed by intraepidermal nerve fiber density and sural nerve amplitude, further suggesting a potential role for mitochondrial dysfunction in the neuropathic processes associated with HIV and its treatment (30, 31).
- Line 197-200. The bottom line is that the level of CD RA mtDNA is low or very low. How can this level impact on the function of these nerves, given that heteroplasmy may compensate these deficiencies?
Response: We appreciate the reviewer’s comment. While it is true that the level of CD RA mtDNA is low or very low, it is essential to recognize that even subtle alterations in mitochondrial genome integrity, such as the presence of CD, can have significant implications for cellular function, particularly in tissues with high energy demands like nerves. Mitochondrial DNA damage, as indicated by the CD, has been associated with impaired mitochondrial function and oxidative stress, both of which are implicated in the pathogenesis of neuropathies such as DSP. Furthermore, while heteroplasmy can modify the effects of mitochondrial DNA mutations to some extent, the extent to which compensatory mechanisms can fully restore mitochondrial function and prevent neuropathy remains unclear and likely varies depending on factors such as the severity of the mutation and the metabolic demands of the tissue.
- Regarding the variation of the expression of the ETC proteins, can the authors discuss whether a postmortem sampling could explain, or partly explain these variations?
Response: We appreciate the reviewer’s insightful comment regarding the potential influence of postmortem sampling on the observed variation in the expression of ETC proteins in our study. While we acknowledge that postmortem sampling procedures could contribute to variations in protein levels, we made efforts to minimize such variations through standardized protocols and careful tissue handling, it is essential to recognize that postmortem factors, such as agonal state, postmortem interval, and tissue preservation methods, can still impact protein expression.
Nevertheless, we agree that discussing the potential impact of postmortem sampling on our results is important for a comprehensive interpretation of our findings. However, studying some human tissues with specific conditions can only be obtained through autopsy.
Additionally, the impact of antiretroviral therapy, particularly nucleoside reverse transcriptase inhibitors (NRTIs), on ETC protein activity further complicates the interpretation of our findings. NRTIs are known to affect specific complexes within the electron transport chain, which may contribute to the observed alterations in protein levels.
- Line 243. The limitation section rather suggests future studies.
Response: We appreciate the reviewer’s comment. We made them 2 separate paragraphs as below:
Page 7, lines 229-240: Our study has several noteworthy limitations. The relatively small sample size due to limited access to autopsied DRG and the SuN specimens may affect the generalizability of our findings. We aim to complete future studies with a larger number of specimens from a diverse group of decedents. While the analyses of biochemical land molecular of DRG and SuN are novel in the context of DSP in PWH, histopathologic assessments of the specimens are needed to confirm that neuropathological changes are consistent with the protein and mtDNA alterations described here in this study. It is important to include such assessments in future studies. The PMI for the specimens included in this study are higher than ideal for neuropathological assessments, and effort will be made in the future to examine tissues with PMI under 8 hours. However, the differences in PMI between the groups are not significant, and the changes in mtDNA and protein levels between the groups are not reflective of increased degradation in the group with the larger PMI.
Page 7, lines 241-267: Future research with larger cohorts is warranted to validate and expand upon our results. Additionally, it is not completely known to which extent the virus itself or ART contributes to the pathogenesis of DSP separately. The impact of ART on mitochondrial function varies widely depending on the specific drugs used, their dosage, and individual patient factors (45). Therefore, investigating the functional consequences of the observed molecular alterations and the direct impact of specific ART on mitochondrial function could provide a more comprehensive understanding of the underlying mechanisms. Future studies could also explore the potential use of mitochondrial protective agents or interventions to enhance mitochondrial transport as therapeutic strategies for DSP. Longitudinal studies tracking mitochondrial dynamics in PWH over time and in response to different antiretroviral regimens could further elucidate the progressive nature of mitochondrial dysfunction in the context of HIV. Finally, as the risk of mitochondrial toxicity from newer ART regimens decreases, the more significant mitochondrial impacts in PWH moving forward may be direct viral toxicity in combination with metabolic disease. Future studies could also explore the potential use of mitochondrial protective agents or interventions to enhance mitochondrial transport as therapeutic strategies for DSP. Targeting pathways that facilitate the transport of mitochondria to distal nerve endings may represent a therapeutic approach to address the specific neuropathic symptoms observed in DSP. Longitudinal studies tracking mitochondrial dynamics in PWH over time and in response to different antiretroviral regimens could further elucidate the progressive nature of mitochondrial dysfunction in the context of HIV. The regional differences in ETC protein expression identified here point to new avenues for research into the specific mechanisms by which mitochondrial dysfunction contributes to sensory neuropathy and may lead to identifying novel drug targets for intervention. Additionally, research into the potential interplay between mitochondrial health, immune function, and neuroinflammation in the context of DSP could provide valuable insights into the multiple causes of this condition and inform the development of comprehensive treatment strategies.
- Line 335. Please reduce this section to the main message.
Response. We reduced this section by combining it with the appropriate paragraph in the Introduction section based on the reviewer’s suggestion. Page 2, lines 65-69:
In a noteworthy prior study of 67 HIV-positive individuals on ART, there was an inverse correlation between mtDNA deletions and peripheral neuropathy as assessed by intraepidermal nerve fiber density and sural nerve amplitude, further suggesting a potential role for mitochondrial dysfunction in the neuropathic processes associated with HIV and its treatment (30, 31).
Minor points
- Figure 1 and 2 are not contributive to the understanding of the manuscript. It rather fit in a PhD dissertation.
Response. We appreciate the reviewer’s comment. We removed Figures 1 and 2.
- Line 148. First time use of DRP1 and ACTB
Response. DRP1 has been already expanded on page 2, line 90. ACTB has been already expanded on page 4, line 136.
- Line 312 First time use of RA
Response. RA has been already expanded on page 3, line 113.